# In Vitro Microbiological and Drug Release of Silver/Ibuprofen Loaded Wound Dressing Designed for the Treatment of Chronically Infected Painful Wounds

**DOI:** 10.3390/antibiotics10070805

**Published:** 2021-07-02

**Authors:** Alejandra Mogrovejo-Valdivia, Mickael Maton, Maria Jose Garcia-Fernandez, Nicolas Tabary, Feng Chai, Christel Neut, Bernard Martel, Nicolas Blanchemain

**Affiliations:** 1U1008 Advanced Drug Delivery Systems, Institut National de la Santé et de la Recherche Médicale (INSERM), Centre Hospitalier Régional Universitaire de Lille (CHU Lille), University of Lille, 59000 Lille, France; amvaldivia26@gmail.com (A.M.-V.); mickael.maton@univ-lille.fr (M.M.); maria-jose.garcia-fernandez@univ-lille.fr (M.J.G.-F.); feng.hildebrand@univ-lille.fr (F.C.); 2UMR 8207, UMET—Unité Matériaux et Transformations, Centre National de la Recherche Scientifique (CNRS), Institut National de la Recherche Agronomique (INRA), Ecole Nationale Supérieure de Chimie de Lille (ENSCL), University of Lille, 59655 Lille, France; nicolas.tabary@univ-lille.fr (N.T.); bernard.martel@univ-lille.fr (B.M.); 3U1286 INFINITE—Institute for Translational Research in Inflammation, Institut National de la Santé et de la Recherche Médicale (INSERM), Centre Hospitalier Régional Universitaire de Lille (CHU Lille), University of Lille, 59000 Lille, France; christel.neut@univ-lille.fr

**Keywords:** wound dressing, antibacterial, anti-inflammatory, drug delivery system, layer by layer, cyclodextrins

## Abstract

This study consisted of developing a dressing loaded with silver (Ag) and ibuprofen (IBU) that provides a dual therapy, antibacterial and antalgic, intended for infected painful wounds. Therefore, non-woven polyethyleneterephtalate (PET) textiles nonwovens were pre-treated by cyclodextrin crosslinked with citric acid by a pad/dry/cure process. Then, textiles were impregnated in silver solution followed by a thermal treatment and were then coated by Layer-by-Layer (L-*b*-L) deposition of a polyelectrolyte multilayer (PEM) system consisting of anionic water-soluble poly(betacyclodextrin citrate) (PCD) and cationic chitosan. Finally, ibuprofen lysinate (IBU-L) was loaded on the PEM coating. We demonstrated the complexation of IBU with native βCD and PCD by phase solubility diagram and ^1^H NMR. PEM system allowed complete IBU-L release in 6 h in PBS pH 7.4 batch (USP IV). On the other hand, microbiological tests demonstrated that loaded silver induced bacterial reduction of 4 Log_10_ against *S. aureus* and *E. coli* and tests revealed that ibuprofen lysinate loading did not interfere with the antibacterial properties of the dressing.

## 1. Introduction

Chronic wounds represent a major public health problem. It is estimated that, in developed countries, 1 to 2% of the population will suffer a chronic wound during their lifetime [1]. Among chronic wounds, leg ulcers, burns or bedsores present an infectious risk, causing excessive inflammation that delays the healing process associated with severe pain and affect psychologic state of patients. Discomfort due to excessive pain, isolation, stress and depression negatively affect the immune system of patients thus increasing healing time. Hence, it is of first importance to treat chronic wounds at the earliest by using effective therapies [2,3,4].

An option to treating infected wounds is the use of technical dressings to provide specific therapies. Currently on the market, there is a wide range of dressings. They consist of different biomaterials and are sometimes combined with active substances to provide specific therapies depending on the etiology of the wound. Among these dressings, silver-based dressings are widely used to treat wounds that are infected or at risk of infection. Several research works have shown the in vitro antibacterial efficacy of silver for infected wounds treatment [5,6,7,8,9,10,11]. Nevertheless, silver presents toxic effects on keratinocytes and fibroblasts inhibiting their proliferation [12,13,14]. Despite this drawback, clinical studies demonstrated the efficacy of commercial silver-based wound dressings on the treatment of chronic leg ulcers, reporting a 45–56% of wound size reduction after 4 weeks [15,16].

However, the presence of silver in the dressings does not address the problem of excessive inflammation produced by these infected wounds and delaying the healing process. Nonsteroidal anti-inflammatory drugs (NSAIDs) are commonly used to treat inflammation, pain, swelling and fever. Ibuprofen is the most frequently used NSAID drug, it is a non-selective inhibitor of cyclooxygenase (COX-2), an enzyme that is responsible for prostaglandin synthesis [17]. However, the systematic administration of ibuprofen may cause side effects, such as a decrease of renal functions or gastrointestinal bleeding [3]. As a matter of fact, clinical studies carried out by Jorgensen et al., [18] Gottrup et al., [2] and Sibbald et al., [19] demonstrated that the use of dressings containing ibuprofen (Bitain^®^-Ibu, Coloplast A/S, Humlebaek, Denmark) reduces pain significantly, without compromising the wound healing process, due to ibuprofen local release.

In recent years, surface treatment techniques have been developed to improve the performance of wound dressings. Techniques, such as encapsulation, fiber coating, chemical or physical grafting are wide strategies used for the incorporation of active ingredients onto dressings supports. Among these technics, fiber coating with polyelectrolyte multilayer (PEM) is really attractive to obtain a multidrug delivery systems [8,10,20,21,22,23,24]. PEM is based on the alternative deposition of positive and negative polyelectrolytes on a substrate, through electrostatic interactions, resulting in the superposition of self-assembled polyelectrolyte layers [25]. Many types of polyelectrolytes; natural, synthetic or semi-synthetic, can be used to develop PEM systems for drug delivery wound dressings. Maver et al. [21], used alginate and viscose for the release of lidocaine and diclofenac; poly(β-amino ester)/alginate or chondroitin sulfate were used to obtain controlled release properties toward vancomycin and diclofenac [22,24]. Polyelectrolytes complexes formed by chitosan/phosvitin, alginate or hyaluronic acid were employed for the release of antimicrobial peptides and silver nanoparticles [9,26,27,28].

Chitosan (CHT) is a polysaccharide obtained from the deacetylation of chitin extracted from crustacean, yeasts and algae. CHT is composed of D-2-deoxy-2-acetylglucosamine and D-2-deoxy-glucosamine units linked by β (1,4) binding [29]. CHT is of many interest in the biomedical field, particularly for wound healing, due to its biodegradability and biocompatibility character [30], its antimicrobial [29,30,31], moisturizing [30], hemostatic [32], and analgesic [29,30,31,33] properties; which are directly related to its cationic character and the size of the polymer chain [31]. In addition, when CHT is prepared in acidic media, a protonation of amino groups occurs and gives to CHT the character of a positive polyelectrolyte fully adapted for the development of a PEM system [28,34,35]. Because of these features, several CHT-based dressings have been developed in the form of fibers, hydrogels, membranes or scaffolds [30].

Cyclodextrins (CDs) are cyclic oligosaccharides contained glucopyranose units joined together by α-1,4 glycosidic bonds. The most commonly used CDs are α-CD, β-CD and ɣ-CD presenting 6, 7 and 8 glucose units respectively [36]. CDs have a truncated cone-shaped 3D structure; their cavity has a hydrophobic character which allows forming inclusion complexes with a large variety of lipophilic molecules, and works as drug carriers to obtain sustained drug delivery systems [23,37,38,39,40,41,42,43]. Furthermore, CDs can undergo chemical modifications in order to enhance their solubilizing power. Anionic cyclodextrins polymers (PCD) increase still more the solubility of active ingredients because of the cooperative action between CDs cavities and the polymeric network [44].

Our team has worked for several years on the formation of polyelectrolyte complexes between CHT and PCD for the formation of hydrogel, nanofibers and PEM system. The cytocompatibility and hemocompatibility of the combination of these polyelectrolyte complexes has been widely demonstrated [45,46,47]. Recently, our group has reported the development of a wound dressing covered by a PEM system formed by chitosan (CHT) as a positive polyelectrolyte and β-cyclodextrin polymer (PCD) as an anionic polyelectrolyte for the sustained release of silver and thus avoid toxicity on keratinocytes and fibroblasts. The objective of this work was to benefit from the PEM coating as a protective layer that would reduce silver diffusion in the wound without losing its antibacterial activity. We have observed that the PEM coating allowed a sustained release of silver (4% of silver released at 72 h) without affecting the antibacterial activity against *S. aureus* and *E. coli*. [10].

The aim of the present work was to develop a wound dressing containing two active compounds; silver sulfate as an antibacterial agent, and ibuprofen as a pain reliever. This paper focused the ibuprofen performance onto dressings, thanks to cyclodextrins properties to form inclusion complexes with drugs. With this purpose, a PEM system formed by CHT/PCD was applied to provide a sustained release for both drugs. For this, a PET nonwoven fabric was pre-treated by βCD crosslinked with citric acid by a pad/dry/cure process in order to obtain an anionic layer for then load silver on the support. After this, a CHT/PCD system was build-up using the dip-coating method until 21 layers were formed. Finally, ibuprofen was incorporated on the dressing by the soaking method. The interaction between ibuprofen and β-cyclodextrin polymer was assessed by ^1^H NMR and phase solubility diagrams, the in vitro ibuprofen release was analyzed in batch conditions and the antibacterial properties of final dressing was evaluated using *Kirby Bauer* and *Kill time* tests.

## 2. Materials and Methods

### 2.1. Materials and Reagents

A non-woven polyethylene terephthalate textile (PET) was used as a textile support. PET textile (NSN 365) was provided by PGI Nordlys (Bailleul, France). PET textile surface weight is 65 g/m^2^ and thickness is 0.186 mm respectively.

Chitosan (CHT) was purchased from Sigma Aldrich (Saint-Quentin Fallavier, France). CHT presents a low molecular weight (140 kg·mol^−1^), a viscosity of 146 cp at a concentration of 1% *w*/*v* (in acetic acid solution 1% *v*/*v*) and a degree of deacylation (DD) of 77% (according to the supplier).

β-cyclodextrin (βCD) was provided from Roquette (Kleptose^®^, Lestrem, France) and the anionic water-soluble β-cyclodextrin polymer (PCD) was synthetized according the method described by Weltrowski et al. [48]. Briefly, citric acid, sodium hypophosphite and βCDs were solubilized in respective weight ratios of 10 g/3 g/10 g in 100 mL of water. After water removal, the solid mixture was cured at 140 °C during 30 min under vacuum. Water was then added, the suspension was filtered, and then dialyzed for 72 h using 6 to 8 kDa membranes (Spectra Por 1, Spectrumlabs). Finally, PCD was recovered by freeze drying. The polymer, poly(cyclodextrin citrate) contained 50% in weight of β-cyclodextrin, 4 mmol per gram of COOH groups (measured by acid-base titration) and the molecular mass was 20,000 g/mol (measured by Size Exclusion Chromatography combined with multi angle light scattering SEC-MALS).

Silver sulfate was provided from Sigma Aldrich (Saint-Quentin Fallavier, France). Silver sulfate (Ag_2_SO_4_, purity ≥99%) was chosen as antibacterial agent. Ibuprofen was used in this work as an anti-inflammatory drug. Ibuprofen (IBU) was purchased from INRESA (Bartenheim, France), it presents a molecular weight of 206.28 g·mol^−1^ and a water solubility of 21 mg/L at 25 °C. Ibuprofen lysinate (IBU-L) is a racemic form of ibuprofen and lysine salt. IBU-L was provided from BASF France (Levallois-Perret, France), IBU-L presents a molecular weight of 352.48 g·mol^−1^ and a water solubility of 35 g/L. The diagram of solubility and proton NMR were performed with IBU to determine the constant of affinity and the geometry of inclusion complex. Drug sorption onto textile was performed with IBU-L, a soluble form of IBU in order to optimize the loading of ibuprofen onto the textiles.

### 2.2. Development of Antibacterial Dressing

Pre-treatment of PET by pad-dry-cure with β-cyclodextrin and citric acid (Figure 1, Step 1–4)): Surface modification of PET supports was applied preliminarily to silver loading and layer-by-layer (*L-b-L)* deposition. The way for providing anionic character to the nonwoven fibers is by modifying fibers with carboxylate groups issued from the reaction between citric acid (CTR) and β-cyclodextrin (βCD) using the *pad/dry/cure* process as a previously reported by our laboratory [10,37,49,50,51,52].

In order to remove the sizing agent, raw PET textile supports were preliminarily washed three times by Soxhlet extractor with isopropanol (3 h) and rinsed three times with ultrapure water, then dried during 15 min at 90 °C.

Aqueous solution of citric acid as crosslinking agent (CTR, 10% *w*/*v*) and sodium hypophosphite as catalyst (NaH_2_PO_2_, 1% *w*/*v*) were prepared. βCD (10% *w*/*v*) was dissolved in the citric acid solution. Weighed raw PET samples (25 cm × 25 cm) were then padded in the final solution, roll-squeezed (roll speed 1 m/min, 2 bars) (ROACHES, Birstall, England), dried at 90 °C for 15 min and cured at 140 °C in a ventilated oven (Minithermo, Roaches, Birstall, England) for 30 min in the case of the βCD. This curing step provoked the crosslinking reaction between βCD with citric acid. Functionalized textiles were washed with ultrapure water three times for 20 min under ultrasound. Finally, after washing, all textiles samples were dried at 90 °C for 15 min. These thermo-fixed primer layer on PET composed of poly(cyclodextrin citrate) were conventionally named as layer #1 on both supports types named respectively PET-CD. Finally, in order to transform residual carboxylic groups (COOH) carried by citrate crosslinks into carboxylate groups (COO^−^), samples were treated with a sodium carbonate solution (4 g/L) for 15 min then rinsed twice with ultrapure water and finally dried for 15 min at 90 °C.

Silver sulfate loading (Figure 1, step 5–6): Functionalized textiles were cut in samples of 5 × 5 cm^2^. Silver sulfate (10 g/L) was dissolved at 70 °C under stirring (80 rpm) for 45 min and filtered with a polyethersulfone filter (0.22 μm, Millex^®^GP). Then, textile samples were impregnated in 65 mL of this solution overnight at 37 °C under stirring (80 rpm). Samples were rinsed with ultrapure water in an ultrasonic bath for 1 min and then left under stirring (80 rpm) for 4 min at 37 °C for three times. Samples were dried at 90 °C for 15 min. Finally, a thermal treatment was applied at 140 °C for 105 min to fixe silver into textiles.

Polyelectrolyte multilayer build-up (Figure 1, step 7–8): The PEM system was built according to the method reported by Decher [25,53] using the dip-coating technique previously described by our group [10,23,49,50]. Samples were firstly dipped into 40 mL of a CHT solution (0.5% *w*/*v*) solubilized in acetic acid (1% *v*/*v*) for 1 min, dried at 90 °C for 15 min, rinsed in acetic acid (0.3% *v*/*v*) to remove excess of chitosan and finally dried at 90 °C for 15 min. Then, the samples were dipped into β-cyclodextrin polymer (PCD) aqueous solution (0.3% *w*/*v*) for 1 min, dried at 90 °C for 15 min, rinsed with ultrapure water and dried at 90 °C for 15 min. The next layers were deposited by applying these sequence 21 times, resulting in overlaying 21 bilayers [10]. A treatment at 140 °C for 105 min was finally applied for the stabilization of the PEM system. Finally, a last step consist on the impregnation in IBU-L solution to load the PEM. The final dressing samples were named PET-CD-Ag-PEM-IBU (Figure 1, step 9).

### 2.3. Ibuprofen-PCD Complexation Study

#### 2.3.1. Phase Solubility Diagram of Ibuprofen

The IBU solubility study was developed according the method described by Higuchi and Connors [54]. A buffer solution (KCl 0.2M and HCl 0.1M) at pH 2 was used to prepared solutions of βCD and PCD at different concentrations (2 mM to 10 mM and 2mM to 44 mM respectively). A volume of 5 mL of these solutions was put into flasks containing 100 ± 5 mg of IBU in order to obtain an excessive concentration of 20 g/L. Then, flasks were kept under stirring (210 rpm, room temperature) for 24 h. Afterward, the samples were filtered through a 0.45 μm cellulose membranes and analyzed by High-Performance Liquid Chromatography coupled to UV detection (HPLC/UV-Vis) (Shimadzu LC-2010A-HT, Shimadzu, Japan). The parameters used were a C18 column (3 × 150 mm), a mobile phase composed of acetonitrile and H_3_PO_4_ at pH 2.25 (40:60 *v*/*v*) a flow rate of 1mL and an injected volume of 20 mL. The detector was set at 225 nm with a retention time of 6.5 min.

The formation constant (K_f_) and the complexation efficiency (CE) values of CD/IBU and PCD/IBU complexes were estimated from the slope of phase solubility diagrams in the initial linear range following to the equations below:(1)$$K_{f}=\frac{slope}{S_{0}\;\left(1-slope\right)}$$
(2)$$CE=\frac{slope}{1-slope}$$
where *S*_0_ is the intrinsic solubility of IBU, the solubility when cyclodextrin is not present.

#### 2.3.2. Proton NMR Spectroscopy

^1^H NMR was performed to reveal the complexation between ibuprofen (IBU) and PCD. For this, IBU-PCD complex was prepared in ultrapure water in equimolar amount (25 mM: 25 mM). The complex was then stirred (100 rpm) at 37 °C for 72 h to obtain the equilibrium and then filtered through 0.45 μm Nylon^®^ membranes, frozen and lyophilized. Finally, 10 mg of the white powder obtained was dissolved in 1 mL of D_2_O to be analyze using a 1H NMR spectrometer Bruker AC300 high resolution 300 MHz.

Two-dimensional NOESY (Nuclear Overhauser Effect Spectroscopy) experiments were operated at 300 K using the standard Bruker parameters and a spin-lock mixing time of 350 ms with TPPI

Method: 2D spectrum consisted of a matrix of 2048 (F2) by 2048 (F1) covering a sweep width of 1929 Hz and 16 increments were collected with 256 transients.

### 2.4. Drug Sorption and Drug Release

IBU-L sorption/desorption and kinetics release were obtained from wound dressings samples disks (Ø 11 mm). Solutions of IBU-L were prepared at different concentrations in phosphate buffered saline (PBS) pH 7.4. HPLC/UV-Vis method was used to quantify IBU-L with the same parameters mentioned above (Section 2.3.1).

#### 2.4.1. Ibuprofen Sorption and Isotherms Analyses

IBU-L loading studies were realized by the batch technique. Briefly, dressing disk samples (n = 6) were soaked in ibuprofen lysinate (IBU-L) solution both at 1 g/L, at room temperature at 240 rpm. These samples were named PET-CD-Ag-PEM-IBU. Desorption of loaded IBU-L was made in PBS pH 7.4 for 24 h under stirring respectively. The total amount of desorbed drug from dressings samples was evaluated by HPLC/UV-Vis method previously described.

For isotherms experiments, dressings disk samples were immersed in IBU-L solutions at different concentrations, following a bath ratio of 5 mL for each sample during 24 h with stirring at 240 rpm at room temperature. Disk samples were then rinsed and analyzed by HPLC-UV/vis technique, as mentioned above, for determinate the quantity of drug loaded (*qe*, mg/cm^2^) and the concentration of the solution at the equilibrium (*Ce*, mg/L), calculated from *qe*.

Two adsorption isotherms models are proposed to understand the adsorption mechanism that are obtained from linearly forms. On one hand, Langmuir isotherm refers the adsorption sites with identical energy, a single molecule by adsorption site and no interaction between ibuprofen molecules, the adsorption is carried out in a monolayer.
(3)$$\frac{Ce}{qe}=\frac{1}{K_{L}}+\frac{LCe}{K_{L}}$$
where *K_L_* (L/g) and *αL* are the Langmuir constants to calculate the theoretical monolayer capacity.

On the other hand, Freundlich isotherm refers to ibuprofen adsorption through non-specific interactions (different bonding energies) between the solute and the support on the one hand, and possibly by stacking of solutes on the adsorbent surface.
(4)$$Ln\;qe=\ln K_{f}+\frac{1}{n}\ln Ce$$
where *K_f_* (L/g) is the Freundlich constant and $\frac{1}{n}$ is the relative adsorption capacity characteristic [40].

#### 2.4.2. Ibuprofen Kinetic of Release

IBU-L kinetic release was assessed in batch. Dressing disk samples were placed in a 24-well plate (CytoOne^®^) containing 2 mL of PBS pH 7.4 as a release medium. The plate was then stirred (80 rpm) at 37 °C. Aliquots were prepared at different time intervals (30 min, 2, 4, 6, 24, 48 and 72 h) and then analyzed by HPLC-UV/Vis method previously described. The withdrawn medium was replaced by an equal volume of fresh PBS solution.

### 2.5. Biological Evaluation

Biological tests were performed following the International and European standards (ISO 10993-5/EN 30993-5) with human lung embryonic epithelial cells (L132, ATCC-CCL5). The L132 cells were cultured in the modified minimum essential medium (MEM, Gibco^®^, Life Technology, Carlsbad, CA, USA) supplemented with 10% *v*/*v* of fetal calf serum (Gibco^®^, Life Technology), streptomycin (0.1 g/L), and penicillin (100 IU/mL), at 37 °C in a CO_2_ incubator (CB 150/APT line/Binder, LabExchange, Paris, France) with 5% CO2/95% atmosphere and 100% relative humidity.

Dressing sample (6 cm^2^) were placed in 1 mL of culture medium (MEM) for 24 h at 37 °C under agitation at 80 rpm (Innova40, New Brunswick Scientific, France). In parallel, 4.0 × 10^3^ L132 cells per well were seeded in a 96-well tissue culture plate containing 100 μL of MEM per well. After 24 h, the extraction medium was collected and sterile filtered (0.2 μm, PB Acrodisc^®^; PALL, Saint-Germain-en-Laye, Enfield, CT, USA). The culture medium was removed from the cells and 100 μL/well of the filtrated extraction medium or CCM (negative control), i.e., absence of cytotoxicity, were respectively added to the wells. After 24 h of incubation, the cell viability was measured by the AlamarBlue^®^ assay (ThermoFisher Scientific, Illkirch, France). Briefly, extraction medium was removed from the cells and 200 μL/well of a 10% AlamarBlue^®^ in MEM solution were added to the wells and placed, protected from light, in an incubator for 2 h. Then, 150 μL of the AlamarBlue^®^ solution were recovered from each well and transferred into a flat bottom 96-well plate. Fluorescence was measured at an excitation wavelength of 530 nm and an emission wavelength of 590 nm, on a microplate fluorometer (TwinkleTMLB 970; Berthold Technologies GmbH & Co, Wildbad, Germany). The fluorescence readings were normalized relative to that of the negative controls. The experiments were performed in triplicate.

### 2.6. Microbiological Evaluation

The microbiological evaluation of dressings was determined against *Staphylococcus aureus* (strain CIP224) and *Escherichia coli* (strain K12). Bacterial culture was realized by inoculating a Mueller-Hinton Agar (MHA) slant incubated for one day at 37 °C. A volume of 10 mL of Ringer’s cysteinated medium was added to the bacteria culture, then bacteria were removed them from the slant. The bacterial suspension contains about 1 × 10^9^ CFU (colony forming unit)/mL approximately.

#### 2.6.1. Kirby-Bauer Test

*Kirby-Bauer* tests were performed to evaluate the influence of ibuprofen lysinate on the antibacterial activity of silver. Disk samples (Ø 11 mm) were dipped in 1 mL of PBS, pH 7.4 and were then stirring (80 rpm) at 37 °C for 72 h with a daily change of PBS solution. 18 mL of Mueller-Hinton agar (MHA) were poured in Petri dishes (Ø 9 cm). Then, 0.1 mL of *E. coli* at 1 × 10^4^ CFU/mL was seeded on the agar. Then, at each interval of evaluation, the textiles samples were deposited onto Mueller Hinton agar plates. After 24 h of incubation at 37 °C, the diameter of the inhibition zone is measured and plotted as a function of contact time in PBS.

#### 2.6.2. Kill-Time Test

*Kill-time* test was performed to evaluate the kinetics of the bacterial reduction to determine the antibacterial activity of dressings [11]. Disks samples (Ø 11 mm) were placed into 24 well plates (CytoOne^®^). Then, 200 μL of a bacterial suspension (1 × 10^7^ CFU/mL) were placed on the textile samples and the plate was incubated at 37 °C. At each interval examination (0.5, 2, 4, 6 and 24 h) disk samples were removed from the well and placed in 2 mL of PBS, pH 7.4, treated in an ultrasonic bath for 1 min and vortexed for 30 s to collect the living bacteria. Successive 1/10 dilutions in cysteinated Ringer solution (CR) were made up to 10^−4^ from the recovered bacterial suspension and 0.1 mL of each dilution was seeded onto Mueller-Hinton agar (MHA). The plates were then incubated for 24 h at 37 °C. The number of viable bacteria was counted and expressed in Log CFU mL^−1^.

## 3. Results

### 3.1. Study of the Interactions between IBU and PCD in Solution

Loftsson et al. have abundantly reported the use of pristine cyclodextrins and cyclodextrins derivates as solubility enhancers of many pharmaceutic molecules in water [36,55]. Crosslinked cyclodextrins polymers also present excellent performances as drug carriers. Trotta et al. have reported the good performances and the versatility of their nanosponges [56]. Besides, our team has reported the same properties of our poly(cyclodextrin citrate) toward triclosan [39], ciprofloxacin [40], simvastatin [46], ethoxzolamide [57].

Ibuprofen is a poorly water-soluble drug and its molecular structure makes it a good candidate to form inclusion complexes with cyclodextrins. Figure 2 shows the phase solubility diagram at pH = 2 of IBU in presence of increasing concentrations of βCD in its native form and in its polymerized form, (PCD), considering PCD contains 50% in weight of βCD. According Higuchi and Connors [47], IBU solubility presents an *A_L_* profile in presence of PCD, indicating a linear increase of drug solubility with PCD concentration.

On the other hand, in presence of βCD, IBU solubility reported a *B_S_* profile which refers to an increase of drug solubility up to [βCD] = 5 mmol/L, followed by a plateau showing a limited solubility levelling off at 2 mM. This plateau is due to the low intrinsic solubility of βCD (16 mmol/L in water at 20 °C) that involves the precipitation of the inclusion complex βCD-IBU [36,55]. This result is in accordance with Pereva et al., [58] and di Cagno et al., [59] who reported the same *B_S_* solubility profile of ibuprofen in presence of βCD.

The *A_L_* profile in the whole range of concentration studied is due to the substantially higher solubility of PCD which is far below its limit of solubility (1 g/mL, corresponding to 0.44 mol/L of βCD units in the solution) compared to pristine βCD. Figure 1 displays in presence of PCD corresponding to a concentration of up to 44 mmol/L in βCD units, IBU apparent solubility increases up to 13.0 mmol/L (2680 mg/L) which corresponds to a solubilization factor of 77. Besides, the limit solubility of IBU is 2.0 mmol/L (412 mg/L) in presence of pristine βCD from 5 mmol/L, corresponding to a solubilization factor of 12 (Table 1).

Association constant (*K_f_*) and complexation efficiency (CE) values calculated from the solubility diagram according to Equations (3) and (4) are presented in Table 1. These data are both calculated from the slopes of the linear increasing portions of the solubility diagrams, i.e., at concentration below 5 mmol/L for βCD and for the whole concentration range for PCD. The slopes values are 0.52062 and 0.31308 for βCD and PCD respectively. Based on these values, calculated *K_f_* value are 6200 L/mmol and 2600 L/mmol, and complexation efficiencies are 1.08 and 0.45, respectively. Such feature can be explained by the higher accessibility of CD cavities in pristine βCD compared to PCD. In the latter case, PCD has a hyperbranched structure forming globular objects of 50 nm diameter [60] where CD cavities situated inside the nano objects present lower accessibility compared to those present at their surface.

However, due to the *B_s_* profile the apparent solubility of IBU levels off at 2.0 mmol/L from 5 mmol/L in βCD, while the *A_L_* profile displayed a IBU concentration increasing up to 13.5 mmol/L for a concentration of 44 mmol/L in the polymerized form of βCD. So, the inconvenience of the lower accessibility of CD cavities in the polymerized form compared to the polymerized form is largely compensated by the very high solubility of PCD. Finally, the slope values of obtained curves are 0.52062 and 0.31308 for βCD and PCD respectively. These values are lower than 1 indicating an inclusion complex of type 1:1 [61].

### 3.2. NMR study of the PCD/IBU complex

A proton NMR study was performed in order to evidence the complexation between IBU and PCD. Figure 3 displays the spectrum of poly(cyclodextrin citrate) (PCD), ibuprofen (IBU) and PCD/IBU complex. The ^1^H NMR spectrum of PCD shows the signal of the glucopyranose units of cyclodextrins, H_1_ (5 ppm); H_3_, H_5_, H_6_, H_4_ and H_2_ are located between 3.58 and 3.99 ppm. The methylene groups of the crosslinking agent (CTR) appear between 2.76 and 2.96 ppm. In Figure 3B the signal of aromatic protons of IBU appear as two doublets corresponding to H_9_ and H_5_ at 7.23 and 7.21 ppm, and H_6_ and H_8_ present at 7.15 and 7.18 ppm respectively. The spectrum of PCD/IBU complex displays a downfield shift of IBU aromatic proton signals at 7.20 ppm and 7.05 ppm. Besides, the signals of the internal protons of CD units H_5_ and H_3_ display upfield shifts as observed in Figure 3C. These spectral changes suggest a complexation of the aromatic group of IBU inside the cavities of CD units of PCD.

The PCD/IBU complex was analyzed more in depth by 1H NMR applying a NOESY sequence (Rotating Frame Overhauser Effect Spectroscopy). As observed in Figure 4, the 2D-NOESY NMR spectrum displays the correlation between H_3_ and H_5_ protons of cyclodextrin cavities with aromatic protons of IBU, confirming the inclusion of IBU into βCDs cavities of PCD.

### 3.3. Ibuprofen Lysinate Adsorption and Kinetic of Release

#### 3.3.1. Adsorption Isotherms

Adsorption isotherm analyses were performed to investigate the adsorption mechanisms of ibuprofen lysinate on PET-CD-Ag-PEM dressings. Adsorption phenomenon is characterized by the equilibrium between dissolved and adsorbed forms of the solute in function of the initial concentration of the solution, that informs on the interactions mechanisms between the sorbent and the species in the solution. In this section, ibuprofen lysinate was preferred to ibuprofen acidic form due to the very low intrinsic solubility of ibuprofen discussed above, which is a limiting factor for the loading of the dressings.

Figure 5a displays the amount of loaded ibuprofen lysinate (IBU-L) (*qe*) onto PET-CD-Ag-PEM dressing as a function of IBU-L concentration in the impregnation bath at the equilibrium (*Ce*, mg/L). The amount of IBU-L adsorbed onto dressing increases as a function of IBU-L concentration used for the impregnation. However, the curve presents two parts. First, an increase (around 30 times higher) in the amount of adsorbed IBU-L was observed when the concentration of the impregnating solution increases from 0.1 g/L to 1 g/L. The concentration of IBU-L adsorbed increased from 3.26 ± 0.21 μg/cm^2^ to 93 ± 4 μg/cm^2^ respectively. Secondly, when the concentration of impregnation increased from 1 g/L to 10 g/L, the amount of IBU-L adsorbed was only 1.7 times higher, presenting adsorbed concentrations of 93 ± 4 μg/cm^2^ and 163 ± 33 μg/cm^2^ respectively.

In the case of ibuprofen acidic form, (data not presented), the same trend as IBU-L was observed. However, the curve was interrupted at 990 mg/L (*Ce*) of IBU-L, because of the solubility limitation in water of this drug in its acidic form.

As shown in Figure 5b, linearization obtained from isotherms data reported an adsorption model similar to the Langmuir model. As a matter of fact, the linearization of *Ce*/*qe* as a function of *Ce*, represents a straight line (R^2^ = 0.9898) and the Langmuir constants were *K_L_*= 163, *qm* = 4.47 µg/cm^2^ and *αL* = 36.48. In contrast, Freundlich model (Figure 5c) showed a straight line (R^2^= 0.9477) lower than the one showed in Langmuir model. Therefore, IBU-L adsorption onto PET-CD-Ag-PEM dressings follows the Langmuir model. This means that drug adsorption is carried out in a monolayer by interaction between one sorption site of the support and one molecule of IBU-L.

Raw data of sorption isotherms (*qe* = *f*(*Ce*) were not possible to exploit accordingly with both Langmuir nor Freundlich models due to the low limit of solubility (<1 g/L) of ibuprofen acidic form. Nevertheless, adsorbed ibuprofen on treated textiles samples loaded by dipping in saturated drug solution reached 95 μg/cm^2^ at equilibrium. Therefore, dressings were loaded with ibuprofen lysinate (IBU-L) in the rest of this work.

#### 3.3.2. Ibuprofen Loading on Dressing Samples

Dressings samples corresponding to PET-CD, PET-CD-PEM and PET-Ag-PEM were impregnated in IBU-L solution at 3 different concentrations: 1, 5 and 10 g/L, for 24 h at room temperature. This assay was carried out to (1) determine the concentration of impregnation bath to obtain an appropriate dose of IBU-L to relieve pain (in comparison to commercial dressing Biatain^®^ Ibu, 500 μg/cm^2^). (2) Evaluate the performance of the PEM system on drug loading, specifically in the presence of silver.

Figure 6 shows the total amount of IBU-L on dressing samples. This result reports a rise of IBU-L adsorption as the concentration of impregnation bath increases, this trend has been observed in the case of isotherms previously reported. PET-CD showed a low charge of IBU-L compared to PET-CD-PEM dressing. The amount of IBU-L adsorbed onto PET-CD was 51, 65 and 105 μg/cm^2^ for a concentration of impregnation of 1, 5 and 10 g/L respectively. At the same time, the amount of IBU-L adsorbed onto PET-CD-PEM was more than double: 109, 173 and 214 μg/cm^2^, respectively. The presence of PCD in the multilayer coating makes possible the ibuprofen lysinate adsorption onto functionalized supports. On the other hand, virgin PET textile was also impregnated in IBU-L solutions with the same concentrations, but the adsorbed amount was not detectable. This situation evidences the importance of PCD for drug loading on dressings supports.

In the case of PET-CD-Ag-PEM-IBU dressings, the results showed a lower amount of IBU-L adsorbed than non-silver loaded dressings, regardless the concentration of impregnation used. This phenomenon is probably due to the fact that silver nanoparticles, interact with amine and carboxylic functions of the thermo-fixed and self-assembled layers of the PEM system, affecting thus the adsorption capacity of the PEM coating.

These results reported that the IBU-L concentration of 10 g/L allowed a maximum drug loading (in the range of 200 µg/cm^2^) on our dressings without reaching the dosage present in commercial dressing (Biatain^®^ Ibu, 500 μg/cm^2^). This concentration was selected to prepare the final dressing named hereafter PET-CD-Ag-PEM-IBU.

#### 3.3.3. Ibuprofen Kinetics Release

Ibuprofen lysinate kinetics release was performed in PBS pH 7.4 and evaluated under static conditions, in order to evaluate the performance of the PEM as drug release system. Figure 7 displays the amount of IBU-L released (μg/cm^2^) as a function of time on different dressings. PET-CD-IBU samples release, after 30 min, around of 80 ± 8 μg/cm^2^, representing a rate of 76% in comparison to the initial dosage, showing a burst effect. Then, the totality of the IBU-L loaded was released in 1 h. These performances reveal that the application of a monolayer (pre-treatment of PET step, layer # 1) is not sufficient to obtain a prolonged release of the drug.

In contrast, samples presenting the PEM coating showed a prolonged release profile. As a matter of fact, PET-CD-PEM-IBU (silver-free dressing) achieves a concentration of 116 ± 5 μg/cm^2^, after 30 min, representing a rate of 54%. Then, the rest of IBU-L loaded (200 ± 15 μg/cm^2^) was released in 6 h. Moreover, the same trend was observed for PET-CD-Ag-PEM-IBU dressing. In fact, the amount of IBU-L released, after 30 min, reached up to 104.0 ± 1.5 μg/cm^2^, representing a rate of 56%. Finally, the total IBU-L loaded (187 ± 5 μg/cm^2^) was also released after 6 h.

The presence of the PEM coating onto textile supports allows a slower release of IBU-L compared to textiles presenting only a monolayer (PET-CD-IBU). Indeed, this is in agreement with our previous works where we observed a prolonged release of silver, chlorhexidine and methylene blue from textile supports coated with similar multilayer assemblies [10,23,49]. These studies displayed that not only the release kinetic was dependent from the number of layers in the layer-by-layer coatings, but was also dependent of a final post-treatment of the modified substrates by thermal or chemical crosslinking of the multi-layered coating.

IBU-L release tests were performed in batch, these results reported a release time over 6 h which can appear very fast. However, the used conditions in this work were different from those of the dressing applied to a wound presenting a very limited exudate flow. Therefore, closer models must be studied. For example, Steffansen and Herping have developed an in vitro release model for the commercial dressing Biatain^®^ Ibu using Franz diffusion cells [62]. The study concludes that the complete release of ibuprofen dose (500 μg) occurs in 7 days with a zero-order kinetics release. The results obtained from this in vitro model were correlated with the data collected from the in vivo release model for dressings proposed by Jorgensen et al., [18]. Thus, the in vitro release model proposed by Steffansen and Herping could be applied to evaluate the drug release profile of IBU-L on dressings developed in this work.

### 3.4. Biological Evaluation

Cell viability was determined on dressing sample by the extraction method, after 24 h of incubation in culture medium (MEM). Figure 8 shows the viability rates of the L132 cells obtained on sample corresponding to each stage of the development of the dressing. The PET exhibited a viability rate of 87%, which shows an absence of cytotoxicity compared to the negative control. The cell viability obtained PET-CD is 79%. PET-CD showed a lower rate of cell viability than PET, which can be explained by the density of residual COOH functions present in textiles provided by poly(cyclodextrin citrate).

On the other hand, the PEM system applied to the samples (PET-CD-PEM) showed optimal cell viability of 101%. It is worth mentioning that the heat treatment (140 °C, 105 min) applied to the PEM system contributed to such good cell viability. As a matter of fact, this treatment provoked the crosslinking of the multilayer assemblies [10] and thus limited the diffusion of PCD in the culture medium preventing the acidification of the cell culture medium (MEM medium enriched with 10% SVF (MEM + SVF)) [63].

However, textiles loaded with silver show cytotoxicity (0% cell viability). This phenomenon is due to the diffusion of part of the silver in the MEM + SVF medium, to reach toxic concentrations. Similar results are obtained with a commercial dressing (UrgoTUL^®^ Ag) under the same analytical conditions. However, with different media (MEM + SVF, MEM and PBS), the amount of silver extracted was 3 to 6 times greater in the commercial dressing than in PET-CD-Ag-PEM-IBU depending on the extraction medium (PBS, MEM, MEM + SVF) (Appendix A). So, this test displayed that the cytotoxicity of our dressing is potentially less toxic than the commercial one as it liberates silver in a lesser extent.

### 3.5. Microbiological Evaluation

Microbiological evaluation was performed in order to evaluate the influence of IBU-L on the antibacterial properties of silver. Figure 9 shows the inhibition zone diameters against *E. coli* for freshly prepared samples and samples after immersion in PBS media (pH 7.4) for 24, 48 and 72 h with the aim of simulating the duration of treatment.

The inhibition diameter presented for PET-CD-Ag dressing was 16.3 ± 1 mm, after 3 days. When the PEM system is applied, the diameter of inhibition of PET-CD-Ag-PEM dressing was 16.3 ± 1 mm at the beginning of analysis and was reduced to 14.6 ± 0.6 mm after 3 days. On the other hand, the dressing containing IBU-L didn’t show significant differences except at T0, probably due to a loss of silver during the impregnation step of dressing in the IBU-L solution overnight. In fact, inhibition diameter at T0 was 14.67 ± 0.58 mm against 16.33 ± 0.58 mm for the dressing without IBU-L to finally reach an inhibition diameter around of 15 ± 1 mm after 3 days, for both kinds of dressings. These results showed that the inhibition diameters are similar between all groups evaluated whatever the analysis time, concluding that IBU-L does not interfere with the silver diffusion through the multilayer assemblies.

*Kill Time* Test (Bacterial reduction test) was developed to evaluate the antibacterial activity of different dressings and to measure the rate at which dressings inhibit the bacterial proliferation.

Figure 10 displays the bacterial reduction (expressed on log CUF/mL) of *S. aureus* (Figure 10a) and *E. coli* (Figure 10b) that were in contact with dressings at different times (0.5 to 24 h). In the case of *S. aureus*, the different dressings show a similar bacterial reduction profile (2.95 log_10_) up to 6 h of evaluation. At 24 h, PET-CD-Ag dressing achieves a bacterial reduction of 6 Log_10_ while PET-CD-Ag-PEM dressing reaches up to 5 Log_10_. On the other hand, the antibacterial activity of dressing with IBU-L displayed a slightly lower bacterial reduction, 4 Log_10_ at 24 h, compared to dressing without IBU-L.

Considering *E. coli* (Figure 10b), the PET-CD-Ag dressing presented a bacterial reduction of 3 Log_10_ at 30 min, following a gradual reduction of bacteria population to finally reach to 4 Log_10_ at 24 h. On the other hand, PET-CD-Ag-PEM dressings showed a faster kinetic reduction. In fact, a bacterial reduction of 4 Log_10_ after 30 min was observed. Then, a reduction of 6 Log_10_ was reached after 6 h of evaluation. Nevertheless, after this time, the bacterial population increased to reach 4 Log_10_ at 24 h. Dressings containing IBU-L showed a slow kinetic reduction compared to dressings without IBU-L. A progressive reduction of the bacterial population until 6 h of contact was perceived, reaching a reduction of 5 Log_10_. Then, the same trend as PET-CD-Ag-PEM was observed. Finally, the bacterial reduction after 24 h reached 4 Log_10_.

Figure 10a,b display that PEM coated samples with dual loading of silver and IBU-L present moderate antibacterial activity compared to samples loaded with silver only within the first six hours of the experiment, however both samples a similar antibacterial activity on the two bacterial strains at 24 h of evaluation.

However, these dressings are still less effective than PET-CD-Ag because they contain the PEM system, preventing the rapid diffusion of silver and thus may reduce toxicity [10].

## 4. Conclusions

In this paper, we reported the development of a wound dressing containing two active ingredients; silver sulfate -as an antibacterial agent-, and ibuprofen—as an antalgic compound—in order to trait infected wounds which chronic pain is associated with. The dressing is based on polyethylene terephthalate (PET) non-woven pre-treated with βCD crosslinked with citric acid by pad/dry/cure technique, loaded then by silver sulfate and finally coated by the layer-by-layer technique with a PEM system made up of chitosan (CHT) as a positive polyelectrolyte and poly(cyclodextrin citrate) (PCD) as a negative polyelectrolyte. In a final step, ibuprofen lysinate was loaded onto dressing samples. Solubility diagram and 1D and 2D proton NMR spectroscopies demonstrated with the inclusion complexation of IBU in CD cavities of PCD. Adsorption isotherms have shown IBU-L adsorption onto multilayer coated samples following the Langmuir model.

PEM coated samples displayed a sustained release of IBU-L up to 6 h in comparison with samples not treated layer-by-layer technique (up to 2 h). Microbiological evaluation revealed that IBU-L loaded multilayer coating keep antibacterial activity against *E. coli* and *S. aureus*.

In the future, in vivo assays in an infected wounded mice model will be developed in order to evaluate the efficacy of wound dressing in infection eradication. Moreover, the in vitro and in vivo therapeutic activity of IBU-L included in this technical dressing will be evaluated in terms of inflammation (IL, TNF, etc.).

## Figures and Tables

**Figure 1 antibiotics-10-00805-f001:**
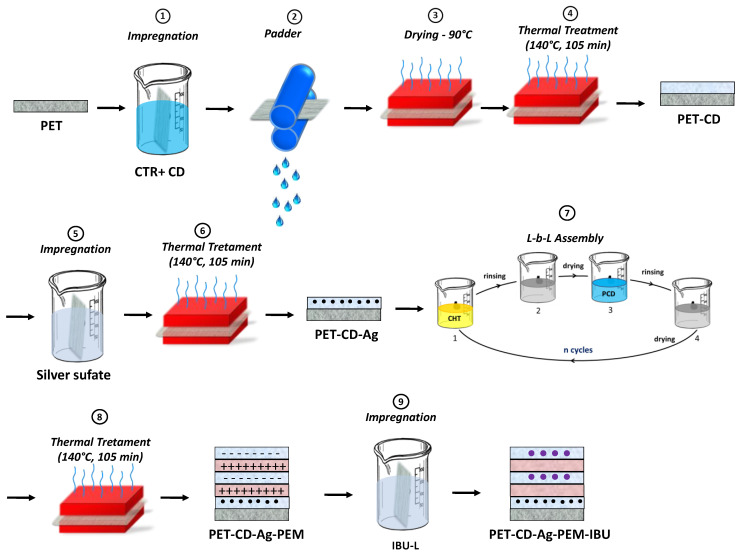
Schematization of preparation of PET-CD-Ag-PEM-IBU.

**Figure 2 antibiotics-10-00805-f002:**
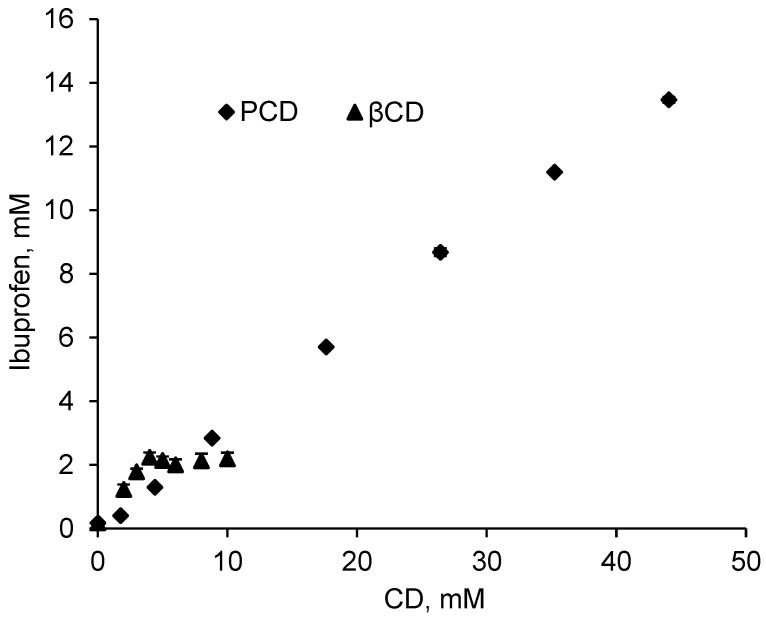
Solubility diagram of IBU with increasing concentrations of pristine βCD and polymerized βCD (PCD) in a pH = 2 buffer solution (KCl 0.2M and HCl 0.1M) at room temperature after 24 h of stirring (210 rpm).

**Figure 3 antibiotics-10-00805-f003:**
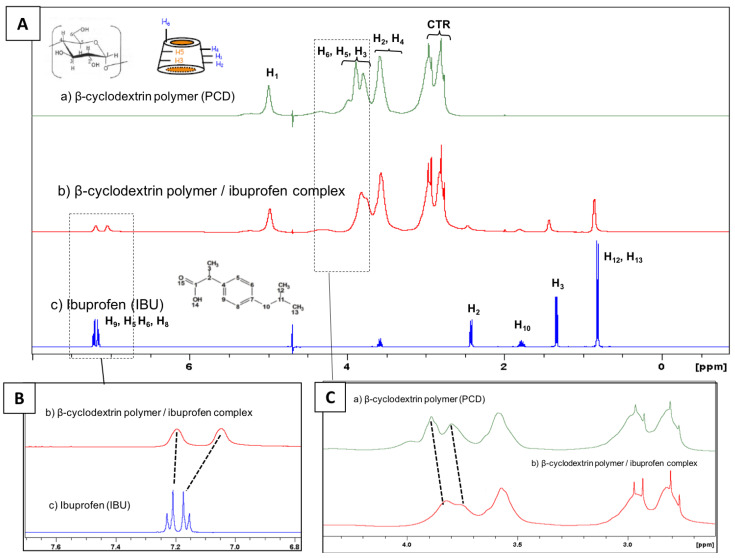
(**A**) Full scale ^1^H NMR in D_2_O of (a) β-cyclodextrin polymer (PCD); (b) β-cyclodextrin polymer/ibuprofen complex and (c) ibuprofen (IBU). (**B**) Magnifications focusing on the signals of aromatic protons of IBU and (**C**) on the signals of PCD protons.

**Figure 4 antibiotics-10-00805-f004:**
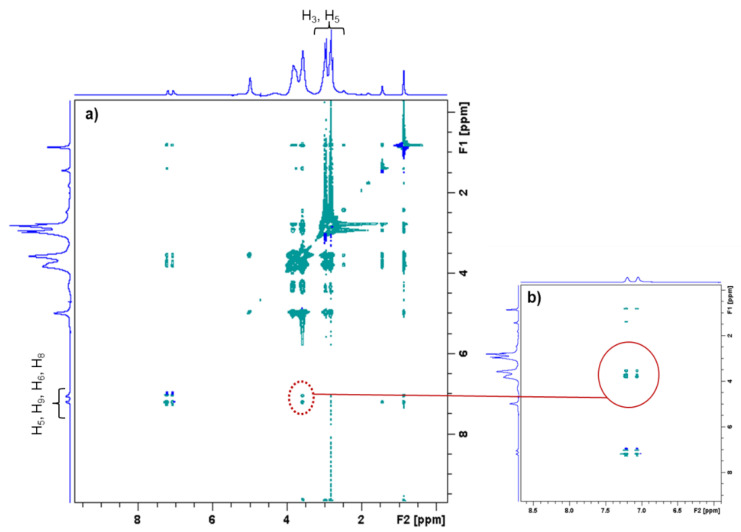
(**a**) 2D-NOESY NMR spectrum of PCD and IBU. (**b**) Enlargement of 2D-NOESY NMR spectrum. The red circle indicates the correlation mark corresponding the inclusion of IBU into the βCD cavities of PCD.

**Figure 5 antibiotics-10-00805-f005:**
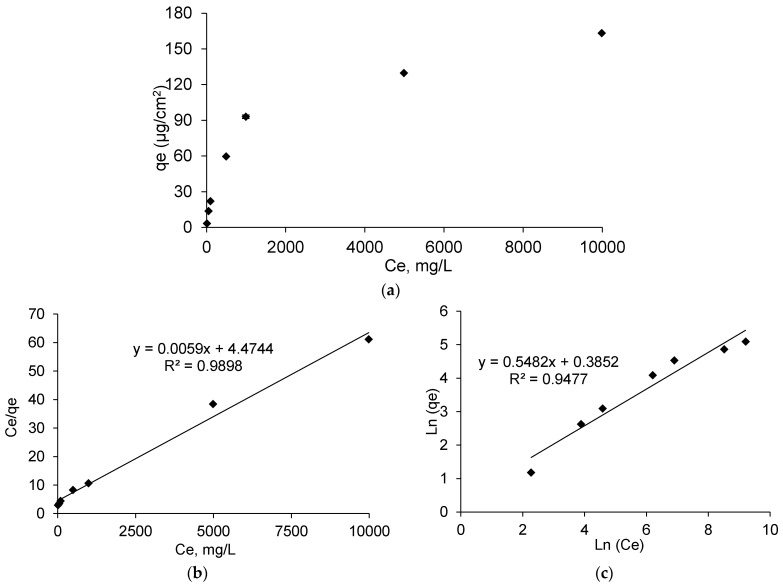
(**a**) Adsorption isotherm of IBU-L onto PET-CD-Ag-PEM dressings after 24 h of impregnation at 240 rpm at room temperature. (**b**) the Langmuir isotherm of adsorption, and (**c**) Freundlich isotherm of adsorption.

**Figure 6 antibiotics-10-00805-f006:**
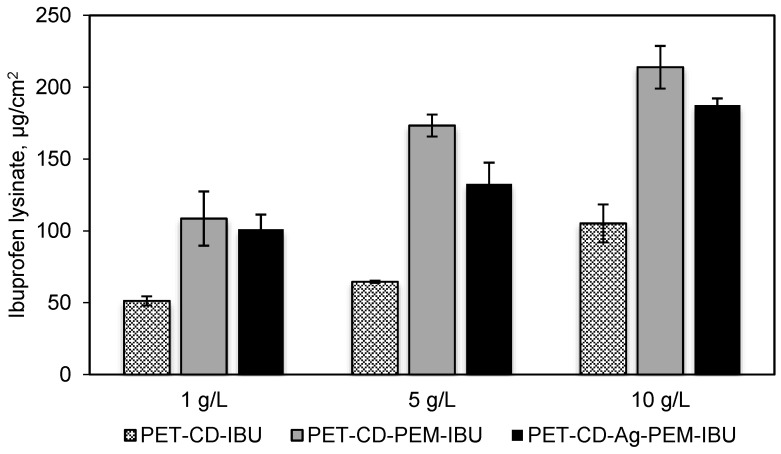
IBU-L adsorbed onto PET-CD, PET-CD-PEM and PET-CD-Ag-PEM dressings. Samples were impregnated in 3 concentrations of IBU-L solutions (1, 5 and 10 g/L) overnight. The assay was carried out by HPLC-UV.

**Figure 7 antibiotics-10-00805-f007:**
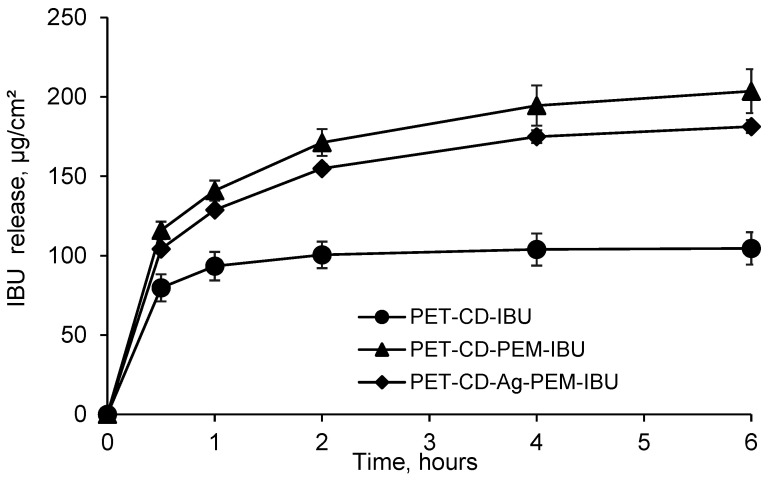
Kinetics release of ibuprofen lysinate (IBU-L) in batch, expressed in μg/cm^2^ (PBS pH 7.4, 37 °C, 80 rpm) from PET-CD-IBU, PET-CD-PEM-IBU and PET-CD-Ag-PEM-IBU dressings.

**Figure 8 antibiotics-10-00805-f008:**
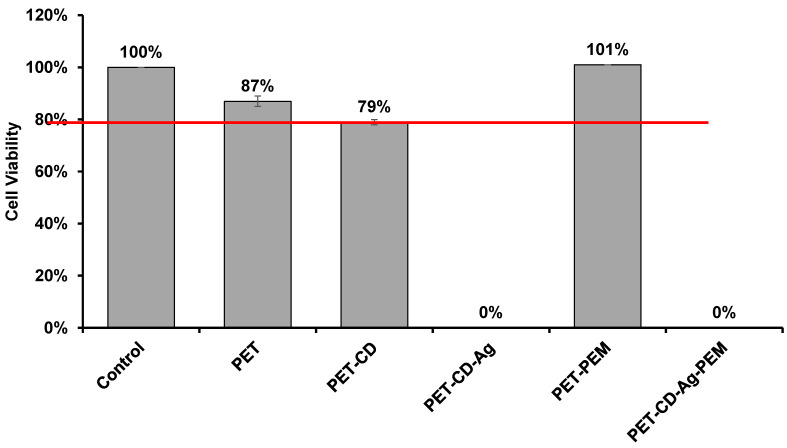
Cell viability of L132 line cell evaluated on different textiles by the extraction method after 24 h of incubation (n = 8).

**Figure 9 antibiotics-10-00805-f009:**
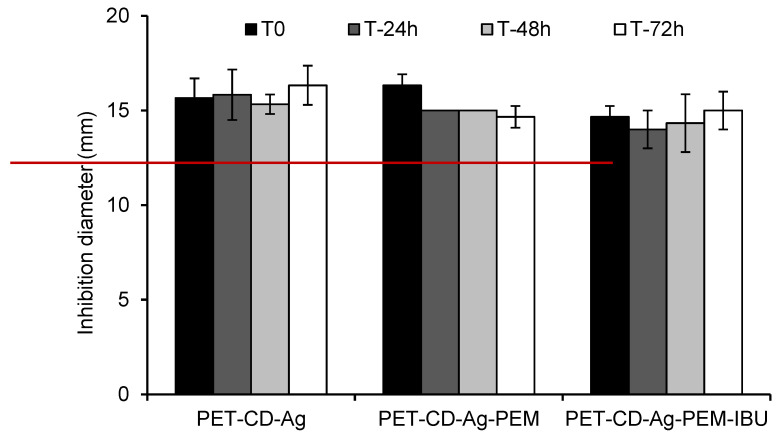
Inhibition diameter in the Kirby–Bauer test against *E. coli* of dressing samples at 0, 24, 48 and 72 h of immersion in PBS pH 7.4 media (80 rpm, 37 °C). The red line in the graph indicates the diameter of dressing samples (11 mm) placed onto Muller-Hinton agar.

**Figure 10 antibiotics-10-00805-f010:**
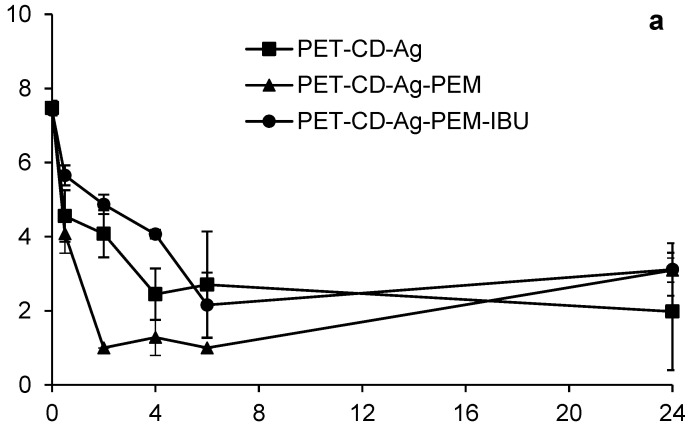
Kinetics bacterial reduction of *S. aureus* (**a**) and *E. coli* (**b**) in contact with different dressings at different times (0.5 to 24 h). Initial bacterial load: 7 Log_10_ CUF/mL.

**Table 1 antibiotics-10-00805-t001:** Association constant (*K_f_*), complexation efficiency (CE) and slope values of βCD and PCD.

	*K_f_* (mM^−1^)	CE	Slope	Solubilization Factor
βCD	6200	1.08	0.52062	12 ^b^
PCD	2600	0.45	0.31308	77 ^a^

Solubilization factor for βCD cavities concentration ^a^ 44 mmol/L, ^b^ 5 mmol/L.

## Data Availability

Data is contained within the article.

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
