# Peer review of "In Vitro Microbiological and Drug Release of Silver/Ibuprofen Loaded Wound Dressing Designed for the Treatment of Chronically Infected Painful Wounds"

_antibiotics, 2021, doi:10.3390/antibiotics10070805_

Round 1
Reviewer 1 Report
Recommendation:
Comments: Publish after minor revisions.
Blanchemain and coworkers report an investigation into PET based wound dressing loaded with silver (Ag) and ibuprofen (IBU). The materials show an improvement of sustained release compared with the materials without applying this layer to layer technique. The preparation and characterization of the materials is thorough and impressive. However, there are some experiments need to be added prior to publication.
- Authors are recommended to evaluate cytoxicity (EC50) of the materials against mammalian cells (e.g. HEK, HeLa etc).
- Authors are recommended to evaluate HC50 of the materials against red blood cells.
- Authors are required to provide 13C NMR diagrams of the compounds. If they are previously reported, please instead reference previous characterization data.
Author Response
Please see in attachment

Reviewer 2 Report
Firstly, the reviewer finds the introduction too way long and would help in readability by reducing necessary context and going straight to the point. Reduction in context from the cyclodextrin section or chitosan section may help make the manuscript more concise.
There is a lack of overall schematic. It is necessary to furbish this manuscript with a graphical illustration as an initial figure to help substantiate this work. Having that as an inset in figure 2A is insufficient.
Much of the analysis is via chemico analysis (NMR) of the material. However, visual characterization and presentation of the film is not made, i.e. SEM or TEM. Even a graphical depiction would help readers to visualize the nature of the material.
While the aims of this paper may be to evaluate the antibiotic nature for the LBL assembly, the reviewer finds this work rather lacking in terms of biological studies. The inclusion of microbial studies in the form of anti-bacterial effect is too rudimentary and there had been very little to account for in terms of inflammation. It may be necessary to examine inflammation reduction in depth via in-vivo/in-vitro IL markers for example instead of just relying on control release studies as shown in figure 6.
What is the tensile strength of this dressing membrane? Nothing of that sort had been presented in this work.
Reviewer 3 Report
The authors evaluated in vitro wound dressing with silver/ibuprofen release for the treatment of chronically infected painful wounds. The topic is of high importance since there is a problem with treatment of hard to heal skin wounds.
I have some comments and questions:
Once to use ibuprofen and once ibuprofen lysinate. What was the reason for it?
In line 460 it is writen that the presence of PEM coating onto textile supports allows a slower release of IBU-L. What is the reason?
There were used appropriate methods however there are missing methods such as hemocompatibility, trombogenicity, wettability.
There are some formal mistakes such as in line 115 there should be were instead of was. Line 179 should be fix. Line 194 there should be dots instead of commas. In lines 339 and in Table 1 there should be dots instead of commas in numbers.
Author Response
The authors evaluated in vitro wound dressing with silver/ibuprofen release for the treatment of chronically infected painful wounds. The topic is of high importance since there is a problem with treatment of hard to heal skin wounds.
I have some comments and questions:
Once to use ibuprofen and once ibuprofen lysinate. What was the reason for it?
IBU in the acid form has low water solubility on contrary of IBU-L which is the basic salt form of ibuprofen (Lysinate). IBU contains only the S stereoisomer, while IBU-L is the racemic mixture. The IBU form was used in the NMR experiment in order to display the inclusion of IBU inside the CD cavity. IBU NMR spectrum displays better compared to IBU-L spectrum.
Using IBU-L in this experiment would prevent this observation due to the additional signals of lysinate on the spectrum, that would superimpose with the CD and IBU peaks.
Besides-IBU-L was used for loading the textile supports: absorption isotherms show that the quantity of IBU loaded onto the samples is dependent on IBU concentration in the impregnation bath. Therefore using IBU would prevent sufficient sorption rates on the dressings to reach the therapeutic effect. Therefore IBU-L was used instead of IBU in this step.
We explain that in Material part “Ibuprofen was used in this work as an anti-inflammatory drug. Ibuprofen (IBU) was purchased from INRESA (Bartenheim, France), it presents a molecular weight of 206,28 g.mol-1 and a water solubility of 21 mg/L at 25°C. Ibuprofen lysinate (IBU-L) is a racemic form of ibuprofen and lysine salt. IBU-L was provided from BASF France (Levallois-Perret, France), IBU-L presents a molecular weight of 352,48 g.mol-1 and a water solubility of 35 g/L. The diagram of solubility and proton NMR were performed with IBU to determine the constant of affinity and the geometry of inclusion complex. Drug sorption onto textile was performed with IBU-L, a soluble form of IBU in order to optimize the loading of ibuprofen onto the textiles.
- In line 460 it is writen that the presence of PEM coating onto textile supports allows a slower release of IBU-L. What is the reason?
The slower release of IBU-L is due to the PEM because it works as a barrier to limit the diffusion of drug as we reported in a previous paper with chlorhexidine. This was specified in the result and discussion section. “The presence of the PEM coating onto textile supports allows a slower release of IBU-L compared to textiles presenting only a monolayer (PET-CD-IBU). Indeed, this is in agreement with our previous works where we observed a prolonged release of silver, chlorhexidine and methylene blue from textile supports coated with similar multilayer assemblies [10, 23, 49]. These studies displayed that not only the release kinetic was dependent from the number of layers in the layer-by-layer coatings, but was also dependent of a final post-treatment of the modified substrates by thermal or chemical crosslinking of the multi-layered coating.”
- There were used appropriate methods however there are missing methods such as hemocompatibility, trombogenicity, wettability.
Indeed, we also could report mechanical tests as requested by reviewer 2 but we prefered to focus on the drug release and antibacterial activity. According to you remark and the remark of reviewer #1 and #4 we added the cytotoxicity test but not the trombogenicity because we already proved in the past the hemocompatibility of polymer vascular devices modified with the same poly(cyclodextrin citrate),and besides, chitosan is already used in many dressings. We added this part in introduction section as ““Our team has worked for several years on the formation of polyelectrolyte complexes between CHT and PCD for the formation of hydrogel, nanofibers and PEM system. The cytocompatibility and hemocompatibility of the combination of these polyelectrolyte complexes has been widely”.
- There are some formal mistakes such as in line 115 there should be were instead of was. Line 179 should be fix. Line 194 there should be dots instead of commas. In lines 339 and in Table 1 there should be dots instead of commas in numbers.
Reviewer 4 Report
The subject of this article is very important and current, in the literature there are several studies on this subject, but the authors use a new approach.
It is known high silver load causes damage, may lead to kidney and liver damages. The cytotoxicity of Ag would be reduced with less Ag load, though the anti-infection effect can be weakened.
This is a very important question, the authors should study the cytotoxicity of the final samples
Author Response
- The subject of this article is very important and current, in the literature, there are several studies on this subject, but the authors use a new approach. It is known high silver load causes damage, may lead to kidney and liver damages. The cytotoxicity of Ag would be reduced with less Ag load, though the anti-infection effect can be weakened. This is a very important question; the authors should study the cytotoxicity of the final samples.
We added the cytotoxicity part in our paper as also asked by Reviewer#1. We clearly observed a cytotoxicity of our final sample but the release less silver than the commercial dressing (with CE and FDA approval).
Round 2
Reviewer 2 Report
Considering that the authors had made changes to the tone on the theme of the paper, the presentation as reflected in this paper is suitable for publication as it stands.
Reviewer 3 Report
I recommend accepting the paper in the present form.